# Biomarkers in Patients with Left Ventricular Assist Device: An Insight on Current Evidence

**DOI:** 10.3390/biom12020334

**Published:** 2022-02-19

**Authors:** Carlotta Sciaccaluga, Nicolò Ghionzoli, Giulia Elena Mandoli, Flavio D’Ascenzi, Marta Focardi, Serafina Valente, Matteo Cameli

**Affiliations:** Department of Medical Biotechnologies, Section of Cardiology, University of Siena, 53100 Siena, Italy; nicologhionzoli@gmail.com (N.G.); giulia_elena@hotmail.it (G.E.M.); flaviodascenzi@gmail.com (F.D.); focardim@unisi.it (M.F.); seravale@gmail.com (S.V.); matteo.cameli@yahoo.com (M.C.)

**Keywords:** LVAD, biomarkers, neurohormonal activation, adverse remodeling, inflammation

## Abstract

Left ventricular assist devices (LVADs) have been representing a cornerstone therapy for patients with end-stage heart failure during the last decades. However, their use induces several pathophysiological modifications which are partially responsible for the complications that typically characterize these patients, such as right ventricular failure, thromboembolic events, as well as bleedings. During the last years, biomarkers involved in the pathways of neurohormonal activation, myocardial injury, adverse remodeling, oxidative stress and systemic inflammation have raised attention. The search and analysis of potential biomarkers in LVAD patients could lead to the identification of a subset of patients with an increased risk of developing these adverse events. This could then promote a closer follow-up as well as therapeutic modifications. Furthermore, it might highlight some new therapeutic pharmacological targets that could lead to improved long-term survival. The aim of this review is to provide current evidence on the role of different biomarkers in patients with LVAD, in particular highlighting their possible implications in clinical practice.

## 1. Introduction

Left ventricular assist devices (LVADs) have been representing a cornerstone therapy for patients with end-stage heart failure (HF) during the last decades, being able to face the growing shortage of heart donations on one hand and on the other hand to serve as an alternative in presence of contraindications to a heart transplant, that to date remains the gold standard [1]. Despite a progressive improvement of LVAD patients’ survival, the medium- and long-term follow-up is jeopardized by a series of complications that have a great impact on prognoses, such as right ventricular failure, bleedings and thromboembolic events [2,3]. Some are determined by natural disease progression (i.e., right ventricular failure), others depend either on the switch of circulation and on compelling drugs used in order to prevent thromboembolic events. Indeed, despite amelioration in materials, these patients still need to be both on effective vitamin K antagonists and on single antiplatelet drugs to avoid surface thrombosis. Even though several hypotheses have been formulated, the exact pathophysiological mechanisms underlying major complications in LVAD recipients are still partially unknown. During the last years, biomarkers involved in the pathways of neurohormonal activation, myocardial injury, adverse remodeling, oxidative stress and systemic inflammation have raised attraction. Despite the extensive studies performed in the setting of HF, their role in patients with LVAD is still far from being clear. Indeed, LVAD profoundly alters the physiology of the whole cardiovascular system since its main function is to draw blood from the left ventricle and pump it back in the ascending aorta (or in rare cases in the descending aorta), by-passing the aortic valve, in order to reduce the pressure and the work of the failing left ventricle. Furthermore, new generation devices overturn normal circulation since they provide a continuous circulation rather than a pulsatile one, leading to several cellular and molecular adaptations. As a consequence, the traditional biomarkers that are well-studied in advanced heart failure might not be suitable for this set of patients, which could potentially lead to misinterpretations in clinical practice. To the best of our knowledge, there is no comprehensive review exploring the role of several biomarkers in LVAD patients and their link to complications development. The aim of this review is to provide current evidence on the role of different biomarkers in patients with LVAD, as exemplified in Figure 1, in particular highlighting their possible implications in clinical practice, especially for what concerns disease progression, early diagnosis of complications and prognostic evaluation.

We included not only the most studied and available markers in LVAD patients, but we also provided an overview of novel and original biomarkers, even though their actual use in clinical practice is still marginal. Intriguingly, as explained further on, there is a notable overlap in deranged mechanisms responsible for disease progression between advanced HF and post-LVAD implantation, despite drivers for their activation may consistently differ.

## 2. Neurohormonal Activation

### 2.1. Sympathetic Nervous System

The first-generation LVADs were characterized by a pulsatile flow, mimicking physiologic circulation. This type of flow was associated with a decreased renin-angiotensin-aldosterone system (RAAS) functioning, reducing, in turn, RAAS-dependent sympathetic nervous system (SNS) activation [4]. Despite being more physiological, these devices had lower durability, required larger external leads (increasing infective risk) and extensive surgical dissection, and were noisier [5].

Continuous-flow LVADs instead revolutionized the clinical management of this cohort of patients, at the price of a deep change in circulation. The continuous flow reduces baroreceptor stimulation, thus, leading to sympathetic stimulation to a greater extent [6]. This mechanism can be further enhanced depending on the residual myocardial activity: if the patient is fully dependent on LVAD-induced circulation, the aortic valve will remain closed, further blunting the excursion of the aortic wall and impairing baroreceptor stimulation [7].

Sympathetic overstimulation during LVADs causes in the early phase an increase in cardiac angiotensin I and angiotensin II levels [4], thus, promoting cardiac SNS activity in a vicious circle. In later phases, chronic overstimulation favors juxta-glomerular and afferent arterial vascular smooth muscle cell proliferation, reducing stretch sensitivity and, in turn, increasing RAAS activity, despite the partial restoration of hemodynamic functions [8]. However, SNS blockade together with LVAD device seems to promote cardiac sympathetic reinnervation, as demonstrated with meta-iodo-benzyl-guanidine scintigraphy in a population of patients with idiopathic dilated cardiomyopathy [9]. Cardiac reinnervation, in turn, has been shown to be a marker of functional recovery after LVAD implantation [10].

Whether these observations affect mortality or the incidence of cardiovascular events, remains mostly unsolved. Little is known also regarding the actual impact of quality of life. A study of cardiopulmonary exercise tests demonstrated an inverse relationship between norepinephrine circulating levels and anaerobic threshold and oxygen pulse, but their levels did not correlate with peak oxygen consumption [11]. Besides, the exercise-induced increase in blood pressure is blunted in patients with LVAD, despite the increase in the pulsatile component related to the intrinsic ventricular activity [12]. LVAD-dependent blood flow is only mildly increased during exercise, as current devices cannot adjust flow depending on peripheral requests [13].

Some biomarkers also seem to predict the clinical outcome before LVAD implantation, especially concerning quality of life. LVAD responders have low levels of β-adrenergic receptor kinase, which would promote internalization/desensitization of β receptors, and high levels of dihydroxyphenylglycol, a catabolite of norepinephrine. High levels of this molecule, with consequent ratio norepinephrine/dihydroxyphenylglycol approximately to 1, point out an active, capable process of catecholamines degradation, and thus, lower exposure [14]. Intriguingly, norepinephrine levels were comparable between responders and non-responders.

### 2.2. Renin-Angiotensin-Aldosterone System

Whilst pulsatile flow from first generation LVAD reduced RAAS activation, new generation, continuous flow LVAD devices cause chronic stimulation of the axis on a mid-to-long-term basis. Immediately after LVAD implantation, blood pressure normalizes and circulating RAAS is diminished. This, in turn, reduces cardiac levels of renin and aldosterone, so that angiotensinogen is no longer rapidly depleted. As a result, myocardial levels of angiotensin I and II are increased, thus activating the SNS. On the other hand, continuous flow induces media wall hypertrophy, blunting mechanical stretch on mechanoreceptors and activating RAAS. Furthermore, myocardial collagen content and cross-linking are increased under LVAD, despite mechanical unloading. Figure 2 shows the interplay between RAAS and the SNS in patients after LVAD implantation.

These phenomena are intuitively mitigated by the use of angiotensin converting enzyme inhibitors (ACEi), including the cross linking of collagen fibers [4,15]. This class of drugs, with or without the use of beta-blockers, was demonstrated to improve survival in a cohort of patients with LVAD [16]. However, the only use of mineralocorticoid receptor antagonists (MRA) did not improve survival, but patients on triple therapy (beta-blockers, ACEi and MRA) showed the best survival rate [17].

### 2.3. Natriuretic Peptide System

An increasing amount of studies is assessing the role of the natriuretic peptide (NP) system in patients with LVAD, including the use of ARNI in this population. These molecules have been studied both in systemic and local myocardial settings. Several studies demonstrated that LVAD implantation reduces circulating levels of N-terminal pro-B-type natriuretic peptide (NT-proBNP) [18], B-type natriuretic peptide (BNP) [19,20] and atrial natriuretic peptide (ANP) [19]. In a retrospective, single-center study of 63 patients, pulmonary capillary wedge pressure was the only independent predictor of NT-proBNP levels after LVAD implantation [21]. Furthermore, in animal models of LVAD, the secretion of ANP in response to an increase in central venous pressure was preserved [22]. However, the increase in central venous pressure leads to higher renal interstitial pressure, with increased renin release. The use of ARNI in the setting of LVAD, favored by the restored hemodynamic functions, reduced NT-proBNP plasma level, without altering serum levels of potassium, creatinine and blood urea nitrogen [23]. N-terminal pro-C-type natriuretic peptide (NT-proCNP) raised progressive attention, as it is mainly produced by endothelial cells and its levels are higher in non-surviving, LVAD-implanted patients. It also positively correlates with a greater inflammatory response [24]. As a consequence, the pro-inflammatory activation of the endothelium may be responsible for a worse clinical outcome, with multi-organ dysfunction. Furthermore, NPs may have a role in the pre-implantation setting for the correct selection of candidates. Pre-LVAD BNP levels predict right ventricular failure >48 h after implantation and correlate with the occurrence of ventricular arrhythmias postoperatively [25], whereas NT-proBNP levels predict early right ventricular failure (<48 h). Both molecules can also predict major adverse events, as well as re-hospitalization until 1.5 years from implantation. Unfortunately, to date, no NP demonstrated to predict all-cause mortality nor left ventricular recovery [26]. NPs can also exert a paracrine/autocrine role. Under physiologic circumstances, the axis of NP/guanylate cyclase (GC)-A/cyclic guanosine monophosphate prevents myocardial fibrosis and hypertrophy [27]. Knock-out mice for BNP develop cardiac fibrosis without an increase in blood pressure [28]. In LVAD recipients, chronic unloading can restore the correct genotype expression, with normalization of ANP/BNP ratio, GC-A/natriuretic peptide receptor-C ratio (17 di observ8) and recovery of GC-A response to ANP [29].

## 3. Markers of Myocardial Fibrosis

As collagen content does not decrease after LVAD implantation, a focus grew on biomarkers of fibrosis. Soluble suppressor of tumorigenicity 2 (sST2) has been widely investigated in HF, both in acute and chronic settings, and despite its use in clinical practice is still missing. In patients with end-stage HF, its levels correlate with the severity of the disease, being higher in INTERMACS I rather than in classes II and III [30]. It holds higher prognostic performance than BNP and NT-proBNP (4 di sST2), and its levels at the time of LVAD surgery [31], as well as interleukin-6 ones [32], have been associated with an increased risk of developing multiorgan failure postoperatively. Circulating levels of sST2 drop after LVAD implantation, with the greatest extent after 1 month and with near-normal levels after 3 months [33].

Galectin-3, a paracrine molecule secreted by macrophages, is directly involved in cardiac fibrosis via TGF-β signaling pathway [34,35]. Despite the number of studies in HF, its role in LVAD recipients in nebulous. First, data found a significant drop immediately after the implantation of the device, as a response to hemodynamic unloading. However, at the time of explant, its levels arise near to pre-implantation values, as if chronic mechanical support would promote inflammation and fibrosis [36,37]. Pre-implantation galectin-3 values may predict the outcome, but the evidence is weak and further studies are demanded [36]. When using NT-proBNP reduction >25% as a surrogate of hemodynamic improvement, galectin-3 was the only biomarker showing a significant reduction.

Finally, persistently elevated markers of extracellular matrix turnover (osteopontin, tissue inhibitors of matrix metalloprotein-1 and matrix metalloprotein-2) [38], together with neurohumoral activation (endothelin-1, NT-proBNP), inflammation (neopterin, procalcitonin) [39] and neutrophil gelatinase-associated lipocalin (NGAL) [40], identify advanced HF patients who will likely develop right ventricular failure after LVAD implantation. Specifically, pre-implantation osteopontin levels >259.2 ng/mL predict the development of right ventricular failure after LVAD implantation [38]. However, many of these markers reflect target organ dysfunction and are not specific to the right ventricle. Figure 3 tries to exemplify the trajectories of neuro-hormonal and fibrotic systems in patients before and after LVAD implantation.

## 4. Endothelial Dysfunction and Neoangiogenesis

The last generation continuous flow LVADs expose the endothelium to increased shear stress which breaks the balance of angiogenetic factors, leading to anomalous vessel development, mucosal hypervascularity and angiodysplasia [41,42]. Therefore, the loss of pulsatility seems to be responsible for gastrointestinal bleeding, which represents one of the main long-term complications in this cohort of patients. In support of this, several in vitro studies demonstrated that the plasma of patients with continuous LVAD shows abnormal levels of angiogenic peptides [41,43], reflecting deregulation in angiogenesis. The underlying mechanisms are still not fully known. However, it has been proven that the anomalous shear stress induces an accelerated degradation of von Willebrand factor (vWF) [44].

vWF is a glycoprotein synthesized by both endothelial cells and megakaryocytes, which is released when a hemorrhage occurs, contributing to the creation of the platelet plug within the vessel. High-molecular-weight multimers of vWF are characterized by a higher hemostatic potential and are degraded by ADAMS-13 in smaller multimers. Several studies reported that the shear stress caused by continuous flow makes the high-molecular-weight multimers of vWF more susceptible to their degradation, leading to increased levels of circulating smaller multimers, causing acquired vWF syndrome, whose incidence is elevated in LVAD patients [45]. Even though the precise effects of elevated levels of circulating vWF are not completely established, the association between acquired vW syndrome, angiodysplasia and bleeding events seems to have a clinical confirmation not only in this set of patients but also in other clinical conditions, such as Heyde syndrome [46]. In fact, even though the hemostatic role of vWF is well-consolidated, there is growing evidence of its involvement in the regulation of several vascular pathways, including angiogenesis [47,48]. One of the most studied pathways is the signaling mediated by vascular endothelial growth factor (VEGF) receptor-2 and angiotensin-2. Indeed, both angiotensin 1 and 2 are molecules implicated in angiogenesis regulation. Angiotensin-1 is synthesized by perivascular cells and it promotes vascular maturity together with normal vessel development. On the other hand, angiotensin-2 is exclusively synthesized by endothelial cells and is stored into Weibel–Palade bodies together with vWF. Once it is released, it competes with angiotensin-1 for tyrosine-kinase Tie2 receptor and, together with VEGF, it promotes a dysregulated angiogenesis characterized by fragile and tortuous vessels [49]. Tabit CE et al. reported elevated circulating levels of angiotensin-2 and a higher expression of angiotensin-2 on endothelial cells in patients with LVAD [41]. This increment might be partially explained by the activation of protease-activated receptor-1 (PAR-1) mediated by thrombin, whose levels tend to be augmented in these patients [50], promoting angiotensin-2 overproduction [41]. In addition to that, Tabit CE et al. highlighted the association between circulating levels of angiotensin-2 and non-surgical bleeding [41].

If on one hand, the loss of pulsatile flow conveys a higher predisposition to bleedings, especially from the gastrointestinal tract, on the other hand, it seems implicated in greater coagulability, leading to a greater tendency for thromboembolic events. In fact, in the presence of continuous flow, end-organ perfusion relies upon both systemic and local regulatory mechanisms. In this context, local mediators play a key role in the endothelium, which in turn are influenced by the characteristics of the blood flow. Non-pulsatile blood flow reduces the release of nitric oxide, one of the principal local vasodilators [51]. Patients with a continuous flow LVAD are characterized by endothelial dysfunction, especially in the microvasculature, which, in turn, triggers an inflammatory response that finally leads to an increased thromboembolic risk.

## 5. Systemic Inflammation

It has been widely demonstrated that the plasmatic levels of inflammatory cytokines, such as tumor-necrosis factor- alfa (TNF-alfa) and interleukin-6 (IL-6), are elevated in chronic HF patients [52,53,54] and they correlate with both the severity of symptoms and the disease, as well as with prognosis [53,54]. In addition to that, in the early stages after LVAD implant, the elevation of inflammation indexes is correlated with higher mortality [55,56]. In support of this, Henning et al. suggested a possible correlation between elevated pre-operative procalcitonin levels with the onset of right ventricular failure post-LVAD implant [39]. Furthermore, Caruso R et al. attested that systemic inflammation, identified by several biomarkers like IL-6, IL-8 and C-reactive protein (CRP), is related to the development of multi-organ failure in the postoperative period [46,47]. The possible mechanisms implicated in the up-regulation of the inflammatory pathways in LVAD patients are still unknown, even though several physiopathological hypotheses have been proposed.

One potential explanation is based on the evidence that when blood encounters a foreign surface, such as the device, it triggers intracellular pathways that finally lead to the release of several chemokines like macrophage inflammatory proteins-1beta (MIP-1beta), granulocyte macrophage-colony stimulating factor (GM-CSF), IL-8, interferon-gamma-induced protein-10 (IP-10) and monocyte chemoattractant protein-1 (MCP-1) [57,58]. Among these biomarkers, a relevant role is played by TNF-alfa. In fact, elevated levels of TNF-alfa are associated with vascular destabilization mediated by pericyte apoptosis together with angiopoietin-1 suppression, giving rise to an augmented risk of bleedings [31]. Indeed, evidence suggests that this cytokine is involved in the pathogenesis of angiodysplasia which is fairly common in these patients [59].

Another explanation is instead based on the fact that both foreign bodies and increased shear stress promote platelet activation and dysfunction which, in turn, favors the formation of platelet-neutrophil conjugates, which are proven to be inflammatory actors in sepsis and ischemia-reperfusion events [60]. Even though CRP is the most commonly used marker of systemic inflammation in clinical practice, its role has been under investigated in the LVAD population. However, some studies reported higher CRP levels after LVAD implant [61] while others attested its negative prognostic role in case of persistently increased values, due to their correlation with organ dysfunction [55].

Finally, a third explanation relies on the change of blood flow from pulsatile to continuous. In fact, in animal models, it was shown that this change implicates a hyperactivation of the renin–angiotensin–aldosterone system [62], which was then confirmed in humans through elevated levels of both aldosterone and plasma renin activity in patients with continuous flow-LVAD compared to previous generation pulsatile flow-LVAD [63]. This implies elevated levels of angiotensin-II and aldosterone and so increased circulating inflammatory cytokines, such as TNF-alfa, IL-6, IL-8 and CRP [64]. Despite these premises, Grosman-Rimon et al. demonstrated how systemic chronic inflammatory persists regardless of the use of neurohormonal blocking agents, suggesting that alternative mechanisms might be involved in the genesis of inflammation [64].

The effective clinical implications of these biomarkers have not been completely elucidated. However, the investigation of the relationship between these continuous flows devices, inflammatory status and the relative cytokines might be particularly relevant since they seem to correlate with myocardial [65,66] and vascular alterations [67,68] potentially adding further elements in the pathophysiology of postoperative complications, as well as potentially providing future therapeutic targets.

As mentioned above, bleeding predisposition is frequently accompanied by thromboembolic susceptibility in LVAD patients. In fact, inflammatory status favors bleeding events due to altered angiogenesis and angiodysplasia development, but it also predisposes patients to thrombotic complications through the inhibition of antithrombotic agent release, such as antithrombin and C protein, creating a procoagulant state [69,70]. Both of them are implicated in regulatory pathways of thrombin genesis, which is over-produced in LVAD patients [50]. Different studies showed that patients with pump thrombosis had higher systemic inflammation status with three to four fold higher CRP levels compared to patients without pump thrombosis [69].

## 6. Conclusions

LVAD patients represent a fragile population, susceptible to several complications both in the short-, medium- and long-term, attributable to pathophysiological modifications induced by continuous flow, to the characteristics of the device and its biocompatibility, as well as the required anti-thrombotic therapy. If on one hand the device allows left ventricular unloading and improvement of end-organ perfusion, on the other, it determines profound alterations in physiological balance. In fact, hemodynamic improvement and potential myocardial recovery are not necessarily parallel with the restoration of a normal molecular and cellular physiology. Indeed, the most relevant complications in terms of prognosis and health burden are right ventricular failure, bleedings and thromboembolic events, that are caused by specific mechanisms with their underlying mediators and molecular pathways. The search and analysis of potential biomarkers in LVAD patients could lead to the identification of a subset of patients with an increased risk of developing these adverse events. This could then promote a closer follow-up as well as therapeutic modifications. Furthermore, it might highlight some new therapeutic pharmacological targets that could lead to improved long-term survival.

## Figures and Tables

**Figure 1 biomolecules-12-00334-f001:**
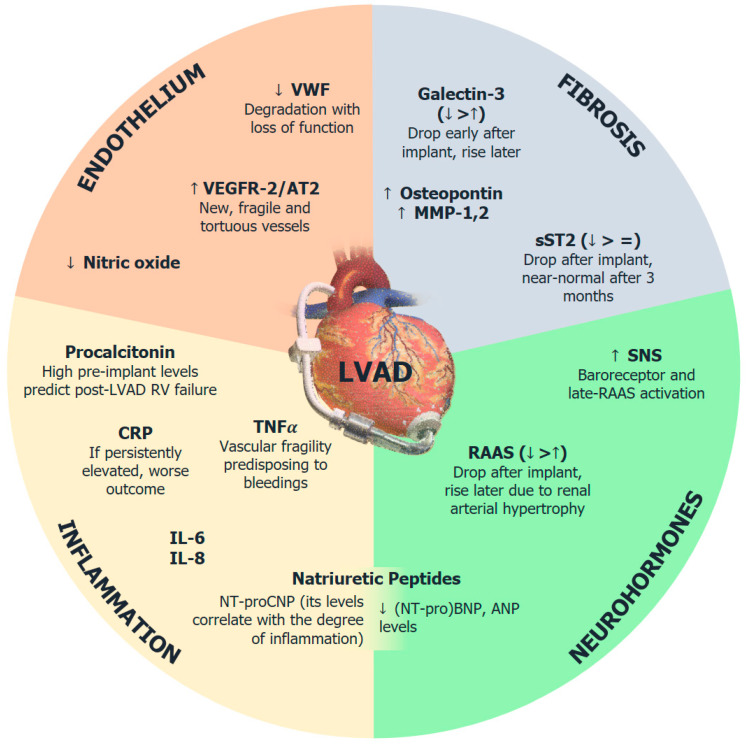
The spectrum of biomarker expression in patients with left ventricular assistance device (LVAD). Four patterns of biomarkers are here reported, with a brief description for each subgroup of the following: endothelium-, fibrosis-, inflammation-related markers and neurohormones. AT2, angiotensin receptor 2; CRP, C-reactive protein; IL, interleukin; LVAD, left ventricular assistance device; NT-proBNP, N-terminal fragment of precursor of B-type natriuretic peptide; NT-proCNP, N-terminal fragment of precursor of C-type natriuretic peptide; RAAS, renin-angiotensin-aldosterone system; RV, right ventricle; SNS, sympathetic nervous system; sST2, soluble suppression of tumorigenesis 2; TNF, tumor necrosis factor; VEGF, vascular endothelial growth factor; vWF, von Willebrand factor.

**Figure 2 biomolecules-12-00334-f002:**
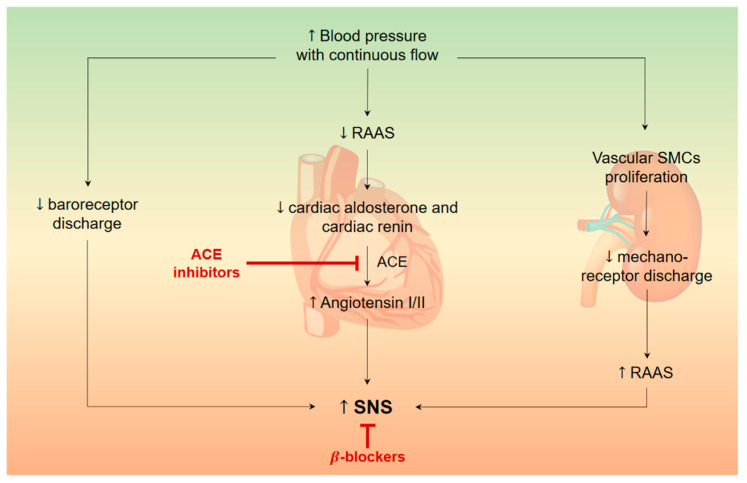
The interplay between renin-angiotensin-aldosterone system (RAAS) and sympathetic nervous system (SNS) in patients after left ventricular assistance device (LVAD). The LVAD-induced continuous flow reduces baroreceptor discharge, thus increasing SNS activity. This is further enhanced by the increase of cardiac angiotensin II levels. Long-term continuous flow induces vascular smooth muscle cells (vSMCs) proliferation with RAAS and further SNS activation. ACE, angiotensin-converting enzyme; RAAS, renin-angiotensin-aldosterone system; SNS, sympathetic nervous system.

**Figure 3 biomolecules-12-00334-f003:**
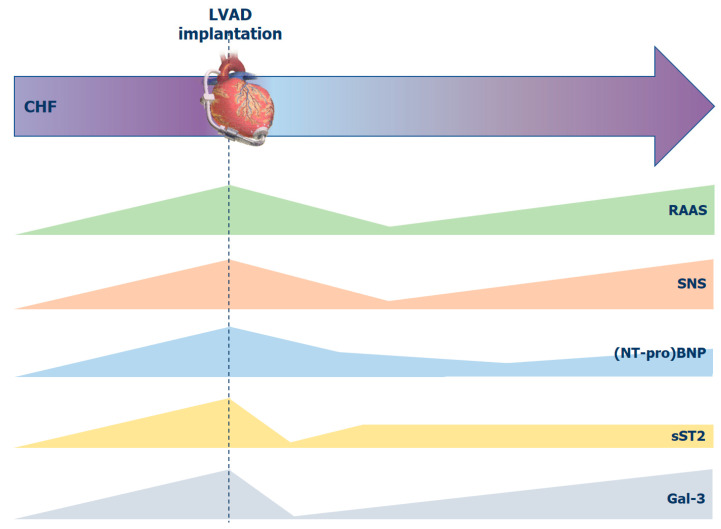
Trajectories of neuro-hormonal and fibrotic systems in patients before and after left ventricular assist device (LVAD) implantation. A figurative timeline is represented above. All markers have the highest levels right before the LVAD implant, but they have different trends afterward. Specific causes for biomarkers elevation/reduction are reported in the text. BNP, B-type natriuretic peptide; CHF, chronic heart failure; Gal-3, galectin-3; LVAD, left ventricular assist device; NT, N-terminal; RAAS, renin-angiotensin-aldosterone system; SNS, sympathetic nervous system; sST2, soluble suppression of tumorigenicity 2.

## Data Availability

Not applicable.

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
