# Peer review of "Biomarkers in Patients with Left Ventricular Assist Device: An Insight on Current Evidence"

_biomolecules, 2022, doi:10.3390/biom12020334_

Round 1

Reviewer 1 Report

Biomarkers in patients with left ventricular assist devices were reviewed in the article from a clinical point of view. A few further research issues were put forward. The article is concise and well-written. However, the article seems to be prepared for the readers who have had knowledge of biomarkers in patients with left ventricular assist devices already. Thus, it is a bit hard to understand the article for the readers who are without that knowledge, especially for biomechanics workers.

The minor issues include that the introduction seems too short, and whether similar articles for the same topic in the literature exist is not mentioned in the introduction.

Author Response

Reviewer#1

Biomarkers in patients with left ventricular assist devices were reviewed in the article from a clinical point of view. A few further research issues were put forward. The article is concise and well-written. However, the article seems to be prepared for the readers who have had knowledge of biomarkers in patients with left ventricular assist devices already. Thus, it is a bit hard to understand the article for the readers who are without that knowledge, especially for biomechanics workers.

Thank you for your comment. As clinicians, we tried to provide a quick overview of the main biomarkers that have been investigated in this setting of patients, even though there are no certain data regarding this field. For this reason, we believe that it important to stress the importance of finding reliable biomarkers in this setting. Indeed, left ventricular assist device is still a field under investigated under many points of view if compared to heart failure. In particular, the pathophysiological changes induced by these devices are still not completely known and therefore the traditional biomarkers used in other settings might be not suitable. On the other hand, it is of paramount importance to find reliable and easy-accessible biomarkers in order to stratify these patients in low-, medium- or high-risk with regards to complications, leading to closer follow-ups and therapeutic changes. Furthermore, it is an interesting field of research for developing potentially new therapies to address the main complications that still have a great impact of LVAD patients’ survival. That being said, we understand that this topic might be particularly challenging even for readers who are familiar with this devices, due to the fact that evidence is still scarce and based on small studies. We tried to provide a quick overview of these devices and how they might affect the physiology of the cardiovascular system and therefore alter the balance which reflects on different biomarker expression, hoping to increase its accessibility to readers.

The minor issues include that the introduction seems too short, and whether similar articles for the same topic in the literature exist is not mentioned in the introduction.

Thank you for your suggestions. We tried to expand our introduction without being repetitive. In particular, we tried to convey the message that the traditional biomarkers used in other clinical context might not applicable to this population due to alteration in cardiovascular physiology. Furthermore, based on your suggestions, we stressed the fact that, to the best of our knowledge, no other comprehensive review of LVAD patients entirely focused of several biomarkers and their link to complications and clinical aspects. 

Reviewer 2 Report

This manuscript gives a nice and comprehensive overview regarding the biomarkers in left ventricular assist devices (LVADs) installed patients. Although LVADs are typic therapy for end-stage heart failure patients, a series of complications are unavoidable, including right ventricular failure, thromboembolic events and bleeding. This review was focused on potential biomarkers in LVAD patients that could help to identify patients with a higher risk of developing adverse events. The spectrum of biomarkers from endothelium, fibrosis, inflammation-related and neurohormones provide detailed and comprehensive information to figure out the unique ones. The specific biomarkers would benefit patients by a closer follow-up and taking therapeutic modification, as well as discovering new pharmacological targets. The structure and expressions are clear and concise. The figures are well-prepared and organized. However, several sessions could be added to provide more value for the reader. 

  1. Inflammation is highly related to various diseases, the biomarkers like TNF-α, IL-6 and IL-8 are also very general. Is there any unique biomarker that is specific or only associated with end-stage heart failure patients? How specific are these biomarkers listed in Figure 1?
  2. For the biomarkers in Figure 1, it’s better to include the trend for them, such as increased or decreased expression levels in patients in LVADs. And how significant are these expressions changed?
  3. A timeline for the biomarker expression should be provided, it would further benefit the researchers and clinical practice.

Author Response

Reviewer#2

This manuscript gives a nice and comprehensive overview regarding the biomarkers in left ventricular assist devices (LVADs) installed patients. Although LVADs are typic therapy for end-stage heart failure patients, a series of complications are unavoidable, including right ventricular failure, thromboembolic events and bleeding. This review was focused on potential biomarkers in LVAD patients that could help to identify patients with a higher risk of developing adverse events. The spectrum of biomarkers from endothelium, fibrosis, inflammation-related and neurohormones provide detailed and comprehensive information to figure out the unique ones. The specific biomarkers would benefit patients by a closer follow-up and taking therapeutic modification, as well as discovering new pharmacological targets. The structure and expressions are clear and concise. The figures are well-prepared and organized. However, several sessions could be added to provide more value for the reader. 

1. Inflammation is highly related to various diseases, the biomarkers like TNF-α, IL-6 and IL-8 are also very general. Is there any unique biomarker that is specific or only associated with end-stage heart failure patients? How specific are these biomarkers listed in Figure 1?

Thank you for the question. Unfortunately, biomarkers in LVAD are still largely unexplored, and as such no specific biomarker was found to predict outcome in patients with LVAD. However, we do think it is important to know and recognise the role of less-specific biomarkers if they might have a role in clinical practice.

2. For the biomarkers in Figure 1, it’s better to include the trend for them, such as increased or decreased expression levels in patients in LVADs. And how significant are these expressions changed?

We really Thank you for the suggestion. We included arrows for the main biomarkers whose clinical trends are known and common in LVAD patients. We decided not to include trends for those biomarkers whose levels are unpredictable, even if their elevation means poorer outcome. E.g.: RAAS elevation is common and pathophysiologically predictable in patients with LVAD; oppositely, despite higher CRP levels predict worse outcome, its elevation is not a rule for patients with advanced HF/LVAD.

3. A timeline for the biomarker expression should be provided, it would further benefit the researchers and clinical practice.

We really appreciated this comment; accordingly, we created a new figure to better visualize time-to-time trends in major neurohormonal and fibrotic pathways, before and after LVAD implantation. We did not include other biomarkers due to the lack of data regarding their temporal expression before and after the implant. We hope this may help the reader to learn biomarkers trajectories.

Round 2

Reviewer 2 Report

The manuscript is apparently improved after adding more information. The authors have well-answered the raised questions. Overall the manuscript is good to go.